# Asian-white disparities in obstetric anal sphincter injury: Protocol for a systematic review and meta-analysis

**Meejin Park**[1], **Susitha Wanigaratne**[2], **Rohan D'Souza**[3,4], **Roxana Geoffrion**[5], **Sarah A. Williams**[6], **Giulia M. Muraca**[3,5,7] *

**1** Faculty of Health Sciences, Department of Global Health, McMaster University, Hamilton, ON, Canada, **2** Sick Kids Research Institute, Edwin S.H. Leong Centre for Healthy Children, Toronto, ON, Canada, **3** Faculty of Health Science, Department of Obstetrics and Gynecology, McMaster University, Hamilton, ON, Canada, **4** Faculty of Health Sciences, Department of Health Research Methods, Evidence and Impact, McMaster University, Hamilton, ON, Canada, **5** Department of Obstetrics and Gynaecology, University of British Columbia, Vancouver, BC, Canada, **6** Department of Anthropology, Brown University, Providence, RI, United States of America, **7** Department of Medicine, Clinical Epidemiology Division, Solna, Karolinska Institutet, Stockholm, Sweden

* muracag@mcmaster.ca

**Data Availability Statement:** No datasets were generated or analysed during the current study. All relevant data from this study will be made available upon study completion.

## Abstract

### Background

Obstetric anal sphincter injury (OASI) describes severe injury to the perineum and perineum and perianal muscles following birth and occurs in 4.4% to 6.0% of vaginal births in Canada. Studies from high-income countries have identified an increased risk of OASI in individuals who identify as Asian race versus those who identify as white. This protocol outlines a systematic review and meta-analysis which aims to determine the incidence of OASI in individuals living in high-income countries who identify as Asian versus those of white race/ethnicity. We hypothesize that the pooled incidence of OASI will be higher in Asian versus white birthing individuals.

### Methods

We will search MEDLINE, OVID, Embase, Emcare and Cochrane databases from inception to 2022 for observational studies using keywords and controlled vocabulary terms related to race, ethnicity and OASI. Two reviewers will follow the Preferred Reporting Items for Systematic Review and Meta-Analysis (PRISMA) guidelines and Meta-analysis of Observational Studies (MOOSE) recommendations. Meta-analysis will be performed using RevMan for dichotomous data using the random effects model and the odds ratio (OR) as effect measure with a 95% confidence interval (CI). Subgroup analysis will be performed based on Asian subgroups (e.g., South Asian, Filipino, Chinese, Japanese individuals). Study quality assessment will be performed using The Joanna Briggs Institute Critical Appraisal tools.

**Funding:** The author(s) received no specific funding for this work.

**Competing interests:** The authors have declared that no competing interests exist.

## Discussion

The systematic review and meta-analysis that this protocol outlines will synthesize the extant literature to better estimate the rates of OASI in Asian and white populations in non-Asian, high-income settings and the relative risk of OASI between these two groups. This systematic summary of the evidence will inform the discrepancy in health outcomes experienced by Asian and white birthing individuals. If these findings suggest a disproportionate burden among Asians, they will be used to advocate for future studies to explore the causal mechanisms underlying this relationship, such as differential care provision, barriers to accessing care, and social and institutional racism. Ultimately, the findings of this review can be used to frame obstetric care guidelines and inform healthcare practices to ensure care that is equitable and accessible to diverse populations.

## Introduction

Racial and ethnic inequities are pervasive in maternal health and are a major source of health disparities in many settings. For example, maternal mortality rates persistently highlight racial disparities in non-Asian, high-income countries such as the United States (US) and the United Kingdom [1,2]. Studies of composite severe maternal morbidity across racial groups in Europe, Australia, and North America have shown similar trends [1,2]. However, the relationship between race/ethnicity and specific causes of maternal morbidities, such as obstetric trauma, is not yet understood.

Obstetric trauma describes severe injury to the perineum, pelvic organs, and supporting myofascial pelvic structures following birth and occurs in 4.4% to 6.0% of vaginal births in Canada [3]. Recent evidence has shown that this rate has been increasing in Canada but reasons for this trend remain speculative [4]. Obstetric trauma contributes to short-term morbidity as well as long-term, life-changing complications such as mental health morbidity, female sexual dysfunction, pain and an increase in most pelvic floor disorders including anal incontinence [5–10].

Evidence is accumulating on the independent association between race/ethnicity and obstetric anal sphincter injury (OASI) in non-Asian, high-income countries such as Australia, Canada, Norway, and the US [11–18]. Although many of these studies have observed higher rates of obstetric trauma among racial and ethnic minorities, these analyses have been limited by poorly defined or inconsistent racial categories, limited sample sizes, and a lack of generalizability. However, a systematic review [19] conducted in 2012 found that rates of OASI among Asian individuals residing in Asian countries were similar to those observed in white, high-income populations. By contrast, those of Asian ethnicity had up to four-fold higher rates of OASI compared with white individuals in Western countries. In the years following this narrative review, researchers' understanding of the quality-of-life impairing long-term consequences of OASI have evolved and resulted in an intensified interest in understanding the distribution of these injuries and their risk factors. Several additions to the literature on this topic have since emerged. Thus, an updated review that includes additional studies since 2012 and synthesizes the evidence on the relationship between Asian race/ethnicity and OASI using meta-analysis is warranted.

The systematic review and meta-analysis outlined in this protocol will include the up-to-date published studies on this topic and aims to summarize these data using meta-analysis to

provide pooled estimates of the association between Asian race and OASI, and investigate any heterogeneity in these associations among different Asian racial/ethnic subgroups. We hypothesize that the pooled incidence of OASI will be higher in Asian versus white individuals.

# Materials and methods

## Study design

This systematic review has been registered in the International Prospective Register of Systematic Reviews (PROSPERO; registration no. CRD42022379141). Preferred Reporting Items for Systematic Review and Meta-Analysis (PRISMA) guidelines [20] and Meta-analysis of Observational Studies (MOOSE) recommendations [21] were used to guide the protocol (S1 File).

## Eligibility criteria

We will search MEDLINE, OVID, Embase, Emcare and Cochrane databases from inception to 2022 for observational studies using keywords and controlled vocabulary terms related to race, ethnicity and OASI. All observational studies, including cross-sectional, case-control, and cohort will be included. Case reports, case series, literature reviews, conference abstracts and gray literature will be excluded from the review. No restrictions based on language will be applied. Studies that do not provide sufficient information to calculate effect size will be excluded.

Studies with all criteria satisfying the PECOS framework will be included in the review.

- Population: Individuals with live or stillbirth in non-Asian, high-income countries.

- Exposure: Asian race/ethnicity will be defined using the outline by the United Nations (UN) [22]. Studies that aggregated individuals of Asian race with other racial groups will not be included. Due to the absence of standard definitions, misuses of the terms 'ethnicity', 'race', and 'ancestry' have been reported [15,23–26]. In this study, we understand the term 'race' to reflect socio-political inequalities, which are often based on "perceived physical differences" such as skin and eye colour [18,23,26]. Ethnicity is defined by cultural factors including language and nationality [26]. For example, there are linguistic, cultural and historical variations within Asian populations [27]. However, discrimination based on a patient's linguistic differences and misinformed cultural assumptions persists [27].

- Comparators: Studies will be included if they compared outcomes in Asian individuals with those in white individuals. The white population serves as a comparison group as they are the predominate ethnic group in non-Asian, high-income countries and because they are not affected by the institutional racism that underlies racial inequities in health. Studies that aggregated white individuals with other racial groups will be excluded.

- Outcome: Obstetric anal sphincter injury (3rd or 4th degree perineal laceration). Third-degree lacerations involve a partial or complete disruption of the internal and/or external anal sphincter. Fourth-degree lacerations involve the disruption of the anal mucosa in addition to laceration of the external and internal anal sphincter [28]. This is the established definition of OASI adopted by the Society for Obstetricians and Gynaecologists of Canada [29], the American College of Obstetricians and Gynecologists [30], and the World Health Organization and the International Consultation on Incontinence [31].

## Search strategy

We will systematically search MEDLINE, OVID, Embase, Emcare, and the Cochrane electronic database from inception to the date of our literature search for observational studies comparing the risk of OASI between Asian and white individuals. We will use keywords and controlled vocabulary terms related to race, ethnicity and OASI, including "race," "ethnicity," "Asian" "obstetric anal sphincter injury" and "severe perineal lacerations." Details of the search strategy are provided in S2 File. Additional papers will be included through hand searching of references of included papers.

## Screening procedure

Two reviewers (MP & GMM) will independently screen titles and abstracts of the articles retrieved from the search for study eligibility. Disagreements will be resolved through discussions with both reviewers. If disagreements persist, conflicts will be raised with the wider study team until consensus is reached. Articles deemed potentially eligible will be carried forward for full-text screening by the two reviewers independently using Covidence (https://www.covidence.org/) to select the final articles using the predefined inclusion and exclusion criteria.

## Data extraction

Two reviewers (MP & GMM) will extract study characteristics including last name of first author, year of publication, country, study design, sample size, overall incidence of OASI, race/ethnicity groups included, Asian subgroups examined, method used to specify race/ethnicity, method used to identify OASI and confounders included in adjusted models (e.g., forceps or vacuum delivery, episiotomy, maternal age, duration of labour, and macrosomic infant). The number of events, number of Asian and white individuals, unadjusted and adjusted odds ratios (OR) as well as 95% confidence intervals (CI) from each study will also be included.

## Risk of bias assessment

Two reviewers (MP, GMM) will independently assess the methodological quality of studies using the Joanna Briggs Institute Critical Appraisal tools, which evaluates risk of bias using a checklist of ten items. These items will be answered with "yes," "no," "maybe" or "not applicable." A numerical score will then be calculated (yes = 1, no/maybe/not applicable = 0). A total score greater than 7 will be considered indicative of low risk of bias, while scores between 4 and 7 will be classified as medium risk, and those between 1 and 3 will be categorized as high risk of bias. Reviewers will resolve any disagreement in bias assessment by discussion. Publication bias will be assessed by constructing funnel plots.

## Data synthesis and analysis

Meta-analysis will be performed using RevMan 5.4 for dichotomous data using the random effects model and the odds ratio (OR) as an effect measure with a 95% confidence interval (CI). We will calculate the unadjusted OR from raw data using the Mantel-Haenszel method. We will also use separate random-effects models to pool the reported unadjusted and adjusted ORs using the inverse variance method. This will enable us to 1) include studies that report unadjusted ORs but do not provide the raw data into a pooled OR estimate of OASI in Asian compared with white individuals and 2) pool adjusted estimates of OASI in Asian compared with white individuals. We will assess the heterogeneity of studies using the $I^2$ statistic. We will

consider heterogeneity significant when the $I^2$ is greater than 50%, following Cochrane Collaboration recommendations.

Subgroup analyses will be performed using studies that compare OASI in specific Asian subgroups (e.g., South Asian, Filipino, Chinese, Japanese individuals) compared with white individuals, by study design (hospital-based versus population-based) and by mode of delivery (operative versus spontaneous vaginal delivery) depending on the number of studies that would allow for sub-analyses.

We will also conduct subgroup analyses excluding studies deemed to have a high risk of bias. In the event that there is high heterogeneity and if there are sufficient studies eligible for synthesis, we will conduct meta-regression to explore the source of heterogeneity among the included studies.

## Discussion

Obstetric trauma is an area of increasing global health importance, yet its differential burden in specific racial/ethnic groups remains understudied. This is of particular concern since a secular increase in the rate of OASI has been evidenced in several high-income countries in recent years [4,32–37] and the impact of this increase on the Asian diaspora is unknown. For example, Asian Americans in the United States have increased in population size by 70% between 2000 and 2020 [38]. Despite this, the health of Asian Americans remains a largely understudied population; only 0.17% of the Institutes of Health (NIH) funding between 1992–2018 was allocated to the subject [38].

Differential access to health care among racial and ethnic minority populations impacts the health outcomes of these populations in majority-white countries. Racism has been a part of medicine and everyday clinical practice for centuries, resulting in quality of life-impairing outcomes and excess morbidity and mortality among BIPOC (Black, Indigenous, and people of colour) individuals [39]. For example, the undertreatment of pain in Black individuals compared with white individuals based on the false biological belief that Black patients experience a lower sensitivity to pain, resulting in inaccurate treatment prescriptions [40].

By answering the question 'is there an increased incidence of OASI in Asian versus white individuals', this systematic review and meta-analysis has many strengths. First, it may confirm that studies from non-Asian, high-income countries indicate an increased risk of OASI in Asian individuals compared with white individuals while displaying lower crude rates of OASI in studies conducted in Asia. Second, it may support the need to advocate for future studies to explore causal mechanisms underlying this relationship. Additional strengths of our project are the proposed subgroup analyses (e.g., by specific Asian ethnicity, Chinese, Japanese, Indian), which has the potential to reveal heterogeneity in the relationship between specific Asian race/ethnicity and OASI. Lastly, this work will serve to advocate for more accurate collection of race-based data (e.g., use of self-report in race/ethnicity measurement) which is critical for advancing health equity. The findings of this study may inform obstetric healthcare practice guidelines on issues related to equitable and accessible care for diverse populations.

The main limitation of this protocol includes challenges in the measurement of race/ethnicity. Dichotomous categorizations of race do not capture the complexity with which race impacts maternal outcomes. We expect to find heterogeneity in the definition of Asian race as has been found in previous research on racial disparities with conventionally used race categories [41]. This can cause misleading results, as it assumes uniform effects within Asian subpopulations and dismisses specific subgroup disparities [38]. In addition, To define Asian countries, we applied a UN classification system named "Standard country or area codes for statistical use (M49)" [22]. While comprehensive, this classification scheme has limitations

because there are several transcontinental countries, such as Kazakhstan, which can be identified as both Asian and European depending on historical, geographical, and cultural contexts [42,43]. This means that the geographical classification used by the UN may differ from the self-identification of race/ethnicity of study participants. We anticipate most studies will include a singular self-reported 'Asian' category that is compared with a singular, self-reported 'white' category. If this is realized, the meta-analysis will be carried out to take a close look at the level of heterogeneity detected.

Any amendments made to the protocol will be reflected on PROSPERO. The manuscript will be submitted to a peer-reviewed journal in the field of obstetrics and gynaecology. Presentation opportunities in research conferences will also be sought.

## Supporting information

**S1 File. PRISMA-P 2015 checklist.** Preferred Reporting Items for Systematic review and Meta-Analysis Protocols 2015 checklist: recommended items to address in a systematic review protocol.
(DOCX)

**S2 File. Systematic review search strategy.** Controlled vocabulary terms related to race, ethnicity and OASI.
(DOCX)

## Author Contributions

**Conceptualization:** Meejin Park, Giulia M. Muraca.

**Data curation:** Meejin Park, Giulia M. Muraca.

**Formal analysis:** Meejin Park, Giulia M. Muraca.

**Investigation:** Meejin Park, Susitha Wanigaratne, Rohan D'Souza, Roxana Geoffrion, Sarah A. Williams, Giulia M. Muraca.

**Methodology:** Meejin Park, Susitha Wanigaratne, Rohan D'Souza, Roxana Geoffrion, Sarah A. Williams, Giulia M. Muraca.

**Project administration:** Meejin Park, Giulia M. Muraca.

**Resources:** Giulia M. Muraca.

**Supervision:** Giulia M. Muraca.

**Validation:** Meejin Park, Giulia M. Muraca.

**Visualization:** Meejin Park, Giulia M. Muraca.

**Writing – original draft:** Meejin Park.

**Writing – review & editing:** Meejin Park, Susitha Wanigaratne, Rohan D'Souza, Roxana Geoffrion, Sarah A. Williams, Giulia M. Muraca.

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
