## [Decision Letter · Decision Letter 0]

28 Jul 2023

PONE-D-23-08342Asian-white disparities in obstetric anal sphincter injury: Protocol for a systematic review and meta-analysisPLOS ONE

Dear Dr. Muraca,

Thank you for submitting your manuscript to PLOS ONE. After careful consideration, we feel that it has merit but does not fully meet PLOS ONE’s publication criteria as it currently stands. Therefore, we invite you to submit a revised version of the manuscript that addresses the points raised during the review process.

Please submit your revised manuscript by Sep 11 2023 11:59PM.  If you will need more time than this to complete your revisions, please reply to this message or contact the journal office at plosone@plos.org. Please include the following items when submitting your revised manuscript:A rebuttal letter that responds to each point raised by the academic editor and reviewer(s). You should upload this letter as a separate file labeled 'Response to Reviewers'.A marked-up copy of your manuscript that highlights changes made to the original version. You should upload this as a separate file labeled 'Revised Manuscript with Track Changes'.An unmarked version of your revised paper without tracked changes. You should upload this as a separate file labeled 'Manuscript'.

We look forward to receiving your revised manuscript.

Kind regards,

Mergan Naidoo, PhD

Academic Editor

PLOS ONE

Reviewers' comments:

Reviewer's Responses to Questions

**Comments to the Author**

1. Does the manuscript provide a valid rationale for the proposed study, with clearly identified and justified research questions?

Reviewer #1: Yes

Reviewer #2: Yes

Reviewer #3: Yes

Reviewer #4: Yes

2. Is the protocol technically sound and planned in a manner that will lead to a meaningful outcome and allow testing the stated hypotheses?

Reviewer #1: Yes

Reviewer #2: Yes

Reviewer #3: Yes

Reviewer #4: Yes

3. Is the methodology feasible and described in sufficient detail to allow the work to be replicable?

Reviewer #1: Yes

Reviewer #2: Yes

Reviewer #3: Yes

Reviewer #4: Yes

4. Have the authors described where all data underlying the findings will be made available when the study is complete?

Reviewer #1: Yes

Reviewer #2: Yes

Reviewer #3: Yes

Reviewer #4: No

5. Is the manuscript presented in an intelligible fashion and written in standard English?

Reviewer #1: Yes

Reviewer #2: Yes

Reviewer #3: Yes

Reviewer #4: Yes

6. Review Comments to the Author

You may also provide optional suggestions and comments to authors that they might find helpful in planning their study.

Reviewer #1: 1- If the female pelvis and perineum are distinct between white and Asian populations, it would be helpful to provide anatomical clues to this difference.

2- are there specific risk factors for OASI among Asians that you intend to monitor?

3- Paragraph 2 in the discussion: "Jumping to give some hints about the relation between the COVID pandemic and racism is not fitting with the context of the topic idea" unless it is well explained.

4. Do you intend to compare the SVD and Instrumental delivery groups and report any differences or ratios? Do Asians have the same elevated risk as people of other ethnicities who have had episiotomies?

Reviewer #2: ABSTRACT: vs should be written in full

INTRODUCTION: There referencing of statements should be sequential, there are no reference numbers 1 and 2.

DISCUSSION: The statement in lines 186-188 should be referenced.

Reviewer #3: A protocol for a systematic study and meta-analysis of the differences between Asian and White women who have obstetric anal sphincter injuries.

The protocol is well written regarding all sections of the study

Reviewer #4: As stated in the title, it does not appear that the group has completed their research that is proposed or perhaps they uploaded the wrong draft? Overall, this study appears to be sound and "will" help answer their hypothesis if carried out properly, but it has not done so yet. Naturally, conducting the research must be done prior to its publication as this protocol and methods are not significantly unique to warrant its own publication.

Further, I encourage the authors to investigate and discuss the question of 'Why?'. Such as why might this trend exist and why does it matter further (which it does).

7. PLOS authors have the option to publish the peer review history of their article (what does this mean?). If published, this will include your full peer review and any attached files.

Reviewer #1: **Yes: **Mena Abdalla

Reviewer #2: No

Reviewer #3: No

Reviewer #4: No

---

## [Author Response · Author response to Decision Letter 0]

4 Aug 2023

Response to Editorial Comments

Comment 1: We note that you have stated that you will provide repository information for your data at acceptance. Should your manuscript be accepted for publication, we will hold it until you provide the relevant accession numbers or DOIs necessary to access your data. If you wish to make changes to your Data Availability statement, please describe these changes in your cover letter and we will update your Data Availability statement to reflect the information you provide.

Response: Since our manuscript describes a protocol for a systematic review and meta-analysis, there are as yet no data related to this project to make available. However, in our revised manuscript, we describe in our Data Availability Statement (lines 24-25) that data we generate will be made publicly accessible when the systematic review and meta-analysis is completed. We have noted this in our cover letter. 

Response to Reviewer Comments

Reviewer 1

Comment 1: If the female pelvis and perineum are distinct between white and Asian populations, it would be helpful to provide anatomical clues to this difference.

Response: As has been shown too often to merit citation, race is a social construct and biological differences within races are more common than between races. Studies on perineal length and its effect on perineal lacerations have found that mean perineal body length does not differ by race.1,2 Rather than pursuing hypotheses based on biological differences between races, we expect to find more meaningful results among studies that interrogate socio-cultural and institutional differences (language barriers, health care delivery). 

References

1. Tsai PJS, Oyama IA, Hiraoka M, Minaglia S, Thomas J, Kaneshiro B. Perineal body length among different racial groups in the first stage of labor. Female Pelvic Med Reconstr Surg. 2012;18(3):165-167. 

2. Yeaton-Massey A, Wong L, Sparks TN, et al. Racial/ethnic variations in perineal length and association with perineal lacerations: a prospective cohort study. J Matern-Fetal Neonatal Med Off J Eur Assoc Perinat Med Fed Asia Ocean Perinat Soc Int Soc Perinat Obstet. 2015;28(3):320-323. 

Comment 2: Are there specific risk factors for OASI among Asians that you intend to monitor?

Response: We will be assessing the eligible studies for whether they considered specific risk factors for OASI such as forceps or vacuum delivery, episiotomy, maternal age, duration of labour, and macrosomic infant. We have included this in the revised manuscript on lines 148-149. 

Comment 3: Paragraph 2 in the discussion: "Jumping to give some hints about the relation between the COVID pandemic and racism is not fitting with the context of the topic idea" unless it is well explained.

Response: We have removed this statement. 

Comment 4: Do you intend to compare the SVD and Instrumental delivery groups and report any differences or ratios? Do Asians have the same elevated risk as people of other ethnicities who have had episiotomies?

Response: We will be evaluating the eligible studies for consideration of risk factors for OASI (lines 148-149). If there are a sufficient number of studies that disaggregate vaginal deliveries by mode of delivery (spontaneous vaginal delivery vs operative vaginal delivery) we will conduct sensitivity analyses of each mode of delivery separately. We have added this information on lines 172-175: “Subgroup analyses will be performed using studies that compare OASI in specific Asian subgroups (e.g., South Asian, Filipino, Chinese, Japanese individuals) compared with white individuals, by study design (hospital-based vs population-based) and by mode of delivery (operative vs spontaneous vaginal delivery) depending on the number of studies that would allow for sub-analyses.”

Reviewer 2 

Comment 1: ABSTRACT: vs should be written in full

Response: This has been corrected (lines 31, 33, 34).

Comment 2: INTRODUCTION: There referencing of statements should be sequential, there are no reference numbers 1 and 2.

Response: Thank you for noting this referencing inconsistency. We have corrected this in the revised manuscript. 

Comment 3: DISCUSSION: The statement in lines 186-188 should be referenced.

Response: We have added the appropriate reference.

“For example, the undertreatment of pain in Black individuals compared with white individuals based on the false biological belief that Black patients experience a lower sensitivity to pain, resulting in inaccurate treatment prescriptions(40).” (lines 191-194)

Reviewer 3

Comment 1: A protocol for a systematic study and meta-analysis of the differences between Asian and White women who have obstetric anal sphincter injuries.

The protocol is well written regarding all sections of the study.

Response: Thank you. 

Reviewer 4

Comment: As stated in the title, it does not appear that the group has completed their research that is proposed or perhaps they uploaded the wrong draft? Overall, this study appears to be sound and "will" help answer their hypothesis if carried out properly, but it has not done so yet. Naturally, conducting the research must be done prior to its publication as this protocol and methods are not significantly unique to warrant its own publication. Further, I encourage the authors to investigate and discuss the question of 'Why?'. Such as why might this trend exist and why does it matter further (which it does).

Response: The manuscript submitted describes our protocol for a systematic review and meta-analysis aiming to determine the incidence of obstetric anal sphincter injury in individuals living in high-income countries who identify as Asian versus those of white race. Our systematic review will explore causal mechanisms assessed in the extant literature

---

## [Editor Report · Decision Letter 1]

24 Aug 2023

Asian-white disparities in obstetric anal sphincter injury: Protocol for a systematic review and meta-analysis

PONE-D-23-08342R1

Dear Dr. Giulia M Muraca

We’re pleased to inform you that your manuscript has been judged scientifically suitable for publication and will be formally accepted for publication once it meets all outstanding technical requirements.

Kind regards,

Mergan Naidoo, PhD

Academic Editor

PLOS ONE
---

## [Editor Report · Acceptance letter]

30 Aug 2023

PONE-D-23-08342R1 

Asian-white disparities in obstetric anal sphincter injury: Protocol for a systematic review and meta-analysis 

Dear Dr. Muraca:

I'm pleased to inform you that your manuscript has been deemed suitable for publication in PLOS ONE. Congratulations! Your manuscript is now with our production department. 

Kind regards, 

on behalf of

Professor Mergan Naidoo 

Academic Editor

PLOS ONE